# Superplasticity in an organic crystal

Satoshi Takamizawa [1], Yuichi Takasaki[1,2], Toshiyuki Sasaki [1] & Noriaki Ozaki[1]

Superplasticity, which enables processing on hard-to-work solids, has been recognized only in metallic solids. While metallic materials and plastics (polymer solids) essentially possess high plastic workability, functional crystalline solids present difficulties in molding. Organic crystals especially are fragile, in the common view, and they are far from the stage of materials development. From the viewpoint of practical application; however, organic crystals are especially attractive because they are composed of ubiquitous elements and often exhibit higher performance than metallic materials. Thus, finding superplastic deformation of organic crystals, especially in a single-crystal-to-single-crystal manner, will pave the way for their material applications. This study confirmed superplasticity in a crystal of a simple organic compound: *N,N*-dimethyl-4-nitroaniline. The crystal exhibits single-crystal-to-single-crystal superplastic deformation without heating. This finding of "organosuperplasticity" will contribute to the future design of functional solids that do not lose their crystalline quality in molding.

[1] Department of Materials System Science, Graduate School of Nanobioscience, Yokohama City University, 22-2 Seto, Kanazawa-ku, Yokohama, Kanagawa 236-0027, Japan. [2] Kanagawa Institute of Industrial Science and Technology, 705-1 Shimoimaizumi, Ebina, Kanagawa 243-0435, Japan. Correspondence and requests for materials should be addressed to S.T. (email: staka@yokohama-cu.ac.jp)

Solids deform by loading force. Deformability is important for the simplification of production processes as well as usage of solid materials. Deformation is classified into two categories: elastic and plastic deformations—the latter is valuable for material processing. A remarkable degree of plastic deformability makes the molding processing quick and easy, and such processing is applied to metals and organic polymers that gain a certain ductility and malleability by heating. However, it has limited application to low-ductility solids, especially those bearing high crystallinity, capable of solid functions based on the structural uniformity of crystalline materials. There is therefore potential demand for another class of plasticity offered by crystalline materials. Up to now, super-plasticity[1–3] has been found as extreme plastic deformability, which drastically improves the workability of hard-to-work materials such as high-strength aluminum alloys[4,5] and titanium alloys[6]. Application of superplasticity to material molding has been developed in the field of metallurgy.

In organic materials, molding processing methods for stiff and hard-to-work crystalline polymers have been developed according to the properties of individual polymers, e.g. Kevlar[7,8]. Common crystalline materials, particularly single crystals, least deserve useful plasticity, in the common view. For this reason, less attention has been paid to the plasticity in organic crystalline solids for a long time. Recently, however, much focus has been placed on U- and S-shaped plastic curving of organic single crystals comprised of aromatic molecules[9–11]. Furthermore, mechanical plastic bending of organic single crystals has been confirmed as one of the classes of diffusion-less plasticity in chemistry, i.e., organoferroelasticity[12–15] and organosuperelasticity[16–19], where superplastic deformability in organic crystals has been missing until now.

In this work, we report the finding of superplasticity in organic single crystals of $N,N$-dimethyl-4-nitroaniline. The crystal deforms over 500% of strain rate along the < 100 > direction based on a multi-layer slipping mechanism while superelastic behavior, or spontaneous shape recovery, is confirmed along the < 201 > direction. It is noteworthy that the crystal keeps single crystallinity during the deformation according to single crystal X-ray diffraction measurements. Actually, the deformed crystal shows superelasticity which is caused by mechanically-induced twinning of a single crystal.

## Results and discussion

**Crystal deformations and crystal structures**. $N,N$-Dimethyl-4-nitroaniline (**1**) (Fig. 1a) is a simple aromatic organic compound. Dimethyl amino and nitro groups are at the para-position of a benzene ring. Compound **1** is known as an organic non-linear optical material due to its large molecular polarizability[20] with a gentle thermal stability ($T_{\text{melting}} = 163\,°C$)[21]. Well-formed yellow plates of **1** ($0.5–5.0 \times 0.2–0.8 \times 0.03–0.10$ mm, $P2_1$ Sohncke group) were obtained by recrystallization from an acetone solution. Photos and crystallographic data of the crystals under various conditions are summarized in Supplementary Note 3 (Supplementary Figures 3–6 and Supplementary Table 1, respectively).

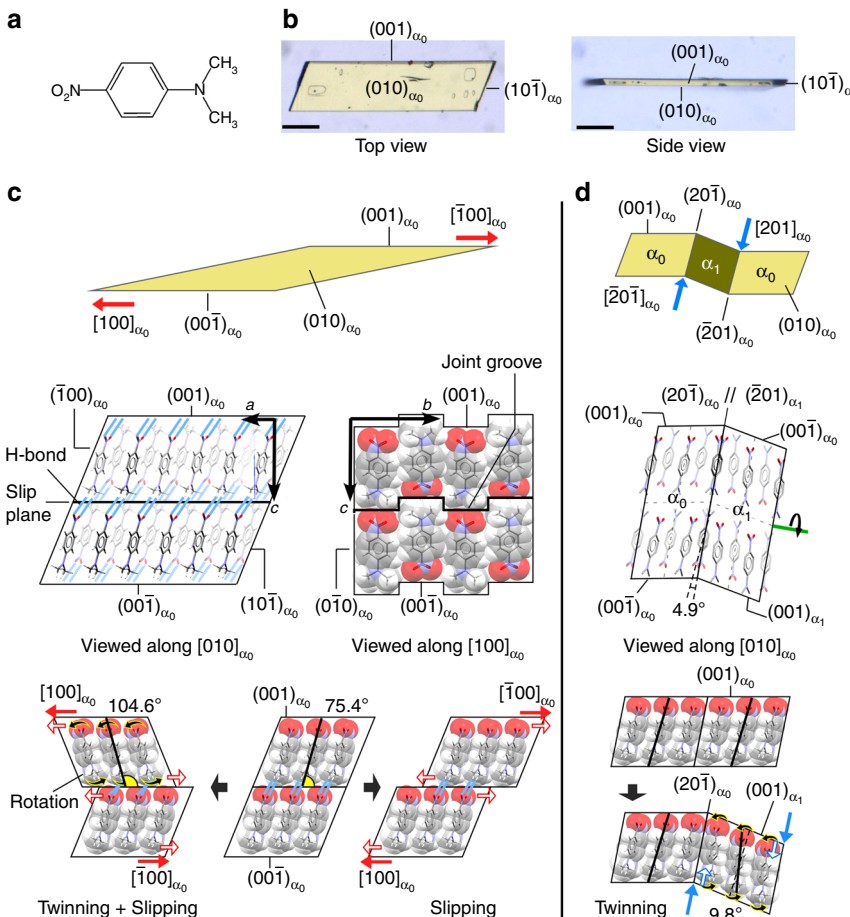

**Fig. 1** Two types of plastic deformation in crystals of **1**. **a** Molecular structure of **1**. **b** Crystal photos of **1** with face indices before deformation. Scale bar, 100 μm. **c** Deformed shape of a crystal of **1** in superplastic deformation, with 500% strain (top), and the micro structures (middle) and slip mechanism showing shearing in the opposite (bottom left) and same (bottom right) directions as the molecular tilting direction in the single crystals. **d** Bending shape (top), micro structure around an interface between twins (middle), and orientation change (bottom) in superelastic deformation

The crystal habit and plastic deformability of crystals of **1** are shown in Fig. 1b–d and Supplementary Note 2. The crystals can plastically curve into an S- or U-shape by stress loading perpendicular to the {001} face (Supplementary Figure 1). Applying shear forces on the (001) and (00$\bar{1}$) faces caused a crystal of **1** to deform, leaning into a parallelogram shape, through multi-layer slipping in the < 100 > direction up to ca. 500% strain (Fig. 1c, Supplementary Movie 1). The slip deformation caused either mere slipping or slipping upon twinning, depending on the slip direction (bottom in Fig. 1c, Supplementary Movie 2). According to X-ray crystal structure analysis, the {001} face and the < 100 > direction were close-packed planes and directions, respectively, forming a slip system (middle left in Fig. 1c). The major axes of the component molecules on the {001} slip face tilted 75.4° against the face, and the molecules formed layers by π-π stacking between benzene rings along the < 100 > direction. The layers then stacked along the < 001 > direction with N = O···H–C hydrogen bonds (*ca.* 3.6 Å from the oxygen to carbon atoms) between the nitro and dimethylamino groups of neighboring molecules. One-dimensional (1-D) grooves, which were parallel to the < 001 > direction and had a 1.4 Å depth and 4.5 Å width on slip planes, engaged one another (middle right in Fig. 1c, Supplementary Figure 7). The engagement restricted layer slipping other than in the < 100 > direction like a "ratchet" and helped to maintain the regulation of interlayer orientation by hydrogen bonds, resulting in anisotropic multi-layer slipping.

There was slip resistance between molecular layers during multi-layer slipping because of contact between the oxygen and hydrogen atoms of the nitro and dimethylamino groups, respectively, which faced each other on slip planes considering their van der Waals radius[22]. When the direction of a shearing vector along the < 100 > direction corresponded with the tilt direction of the major axes of molecules on the {001} slip plane, multi-layer slipping proceeded while keeping molecular orientation (bottom right in Fig. 1c). On the other hand, when the directions were different from each other, twinning deformation occurred by molecular rotation, which was induced by the slip resistance, in addition to slip deformation (bottom left in Fig. 1c, Supplementary Figure 2).

Next, shear stress was applied to a crystal of **1** along the < 201 > direction on the {20–1} face, which intersected the slip plane for superplastic deformation with a tilt angle of 79.8°. The crystal showed superelasticity by twinning deformation. A daughter (twinned) domain ($α_1$) was generated along the shearing direction, resulting in bending of the crystal at the interface of the mother ($α_0$) and $α_1$ domains. The original shape of the crystal was recovered spontaneously by reversion of the $α_1$ domain to the $α_0$ domain after removal of the stress (Fig. 1d, Supplementary Movie 3). X-ray crystal structure analysis of the bent crystal revealed that the interface was $(−201)_{α0}//(20−1)_{α1}$ (or $(20−1)_{α0}//(−201)_{α1}$) and the twinning deformation was caused by a 180° lattice rotation about the axis perpendicular to the interface (green line in Fig. 1d). During the twinning deformation, the major axes of the components in each domain tilted 4.9° against the interface. The spontaneous shape recovery was therefore attributable to frustration of molecular assembly, which was caused by the tilt, at the twinning boundary (Fig. 1d, Supplementary Figure 8).

According to the findings in this investigation, crystals of **1** exhibit superplasticity and superelasticity by slip and twinning deformations, respectively, depending on the shear direction. It is noteworthy that the low symmetry of molecular crystals, which is attributable to the structural anisotropy of organic molecules, leads to anisotropic crystal deformation, i.e. limited directions of slip deformation and selectivity between superplasticity and superelasticity in the present case.

**Stress–strain curves of superplasticity and superelasticity**. The relationship between stress and strain during superplastic deformation in a crystal of **1** was subsequently investigated at room temperature (Fig. 2). Faces (001) and (00–1) of the crystal were attached to glass jigs separately. The faces then slid when the jig on the (001) face was moved along the [100] direction at a constant speed of 50 μm min$^{-1}$ (Fig. 2a). Shear strain rate $ε$ caused by the slip deformation was calculated by dividing displacement magnitude $x$ by crystal height $h$ (338 μm). Shear stress $σ$ was calculated by dividing the observed force by the cross-sectional area ($S_0$: 0.027 mm$^2$) of the original crystal.

By the sliding of the jig, the crystal started to deform plastically by slipping at the yield point defined by the critical shear stress (0.34 MPa, Fig. 2b (0.8%)). In the region of $ε = 0$–325%, the slip deformation proceeded in the whole crystal ($σ = 0.25$–0.34 MPa). The stress in the slip deformation of crystals of **1**, originating

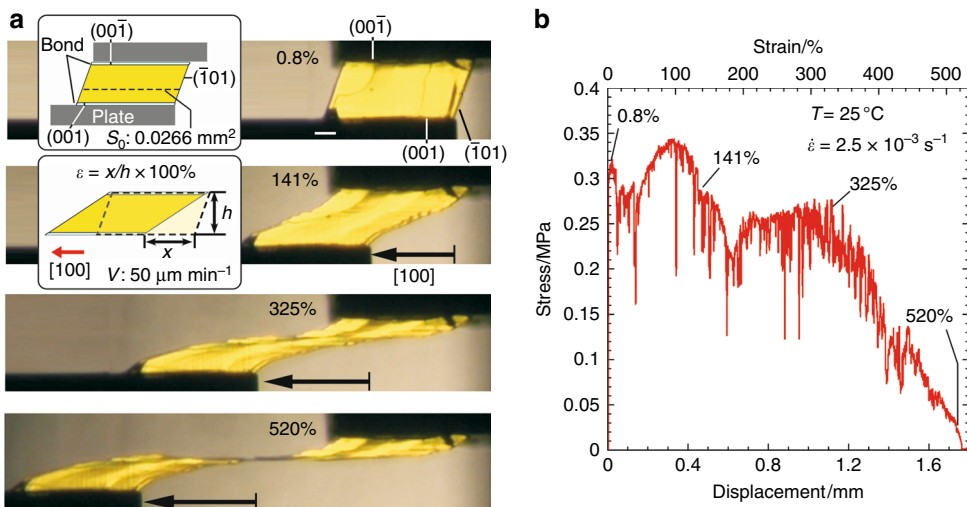

**Fig. 2** Shear test of a crystal of **1**: superplasticity. **a** Snapshots during crystal deformation. Scale bar, 100 μm. **b** Stress–strain curve of **a**. The crystal deformation was observed under a polarizing microscope and measured under the strain rate $\dot{ε}$ of $2.5 \times 10^{-3}$ s$^{-1}$ (0.833 μm s$^{-1}$ / 338 μm) at room temperature

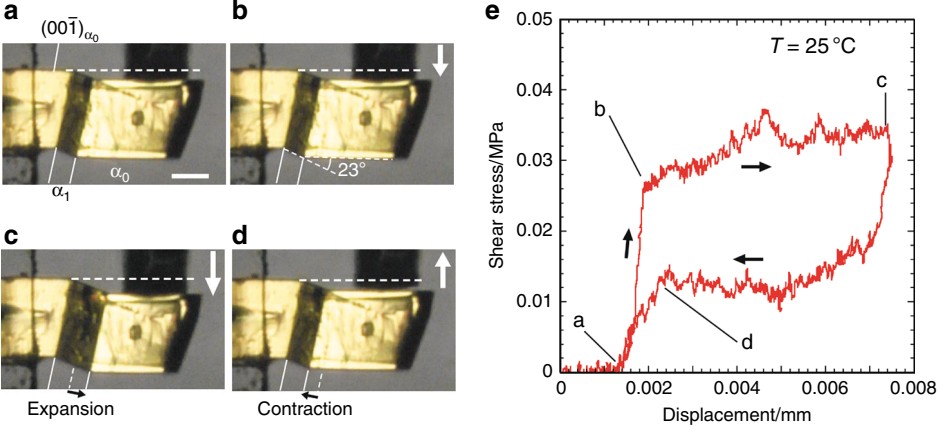

**Fig. 3** Shear test of a crystal of **1**: superelasticity. **a–d** Snapshots of a crystal of **1** during deformation. Scale bar, 100 μm. **e** Stress–strain curve at room temperature. a–d in **e** represent the same states of the crystal presented in **a–d**

from molecular interactions, is remarkably small in comparison to that of superplastic metals, e.g., $Ni_3Al$ alloys (ca. $\sigma = 50$–$300$ MPa)[23]. On one hand, metal materials are hardened by superplastic deformation because of multiplication of dislocations. On the other hand, single crystals of **1** showed no obvious hardening during multi-layer slipping in whole crystals up to $\varepsilon = 325\%$ (fluctuation range of $\sigma = 0.2$–$0.3$ MPa, ± 20%). It means that superplastic deformation in single crystals of **1** proceeded highly idealistically. In the region of $\varepsilon = 325$–$520\%$, the crystal deformed considerably, and the slip deformation proceeded mostly around the middle of the crystal, accompanied by a linear decrease of stress. This result reflects a linear decrease of the slip plane area because of localization of the deformation.

Strain rate sensitivity index $m$, which is the rate of increase of critical stress against the rate of increase of strain ($\dot{\varepsilon} = v/h$), is empirically represented by $\sigma = K\dot{\varepsilon}^m$ ($\sigma$: critical stress, $K$: solid-dependent constant, $\dot{\varepsilon}$: distortion rate)[24–27]. The $m$ values of crystals of **1** (0.036–0.137) are remarkably low compared to those of superplasticity of metals ($m$: 0.3–1.0)[1–3], suggesting a great advantage of crystals of **1** over metals from the viewpoint of rapid superplastic deformation (Supplementary Figures 9 and 10, Supplementary Movies 4 and 5).

Stress–strain cycles of superelastic deformation in a crystal of **1** were then measured at room temperature. The crystal, whose (001) face had been fixed on a base, was sheared when the jig moved along the normal vector of the (00–1) face (Fig. 3a–d). The bending angle between the $(001)_{\alpha0}$ and $(001)_{\alpha1}$ faces was 23°, which was measured based on microscope observation. The value is close to the bending angle of 20.3° calculated based on X-ray crystallographic studies (Fig. 3b). Typical superelastic stress–strain cycles were obtained from the shear test. Crystal bending by twinning deformation was observed at the critical shear stress (0.028 MPa). Subsequent crystal distortion proceeded under a constant stress (0.030–0.032 MPa) with the growth of the $\alpha_1$ domain. The crystal spontaneously recovered its original shape with a constant recovery stress (0.01 MPa) after the shear stress was removed (Fig. 3e, Supplementary Movie 6).

The critical shear stress of a crystal of **1** is less than one-half to one-eighteenth of that of previously reported organosuperelastic crystals[16–18]. In addition, the superelasticity of a crystal of **1** shows a small temperature-dependence (critical shear stress of around 0.030–0.035 MPa in the temperature range 25–50 °C) (Supplementary Figure 11). General superelastic materials show a large temperature-dependence, which means it is difficult to create superelastic materials insensitive to temperature changes[28]. Crystals of **1** are therefore handy superelastic materials working with small and constant forces in a wide range of temperatures.

**Combination of superplasticity and superelasticity.** The superplasticity of crystals of **1** is due to highly anisotropic multi-layer slipping guided by the 1-D grooves on slip planes and regulated by interlayer hydrogen bonds. This means that superplastically deformed crystals can retain single crystallinity and superelasticity. Based on this idea, we conducted shear tests and single crystal X-ray structure analysis on a superplastically deformed crystal of **1** at room temperature. Superelastic behavior was observed in superplastically deformed moieties of the crystal with a shear strain rate of ca. 400% (Fig. 4a, Supplementary Movies 7 and 8). In addition, the retention of single crystallinity of the crystals (deformation rate: 100–150 and 330%) was confirmed by single crystal X-ray diffraction measurements (Supplementary Figures 5 and 6, Supplementary Table 1 and details in the supporting information).

Superplasticity and superelasticity were then coupled in a crystal of **1**. The crystal superplastically deformed during superelastic deformation. The crystal recovered a linear shape by dissipating twins during slip deformation in a single-crystal-to-single-crystal manner (Fig. 4b, Supplementary Movie 9). Multi-layer slipping is an effective mechanism for suppressing transition at an interface between $\alpha_0$ and $\alpha_1$ domains by continuously varying the volume rate between the layers with molecular orientation changed by shearing in the $\alpha_1$ domain and the layers with unchanged molecular orientation (Fig. 4c). This can be regarded as solid-state recrystallization of disarranged microcrystal domains through material deformation.

Superplasticity, which has been known in a limited number of metal solids, was confirmed in single crystals of a simple organic compound—N,N-dimethyl-4-nitroaniline—at room temperature. Molecular crystals are constructed by weaker interactions and are less symmetric than metal solids. Thus superplasticity of molecular crystals makes it possible to conduct plastic processing of solids without losing characteristic solid properties, superelasticity in the present case, under heat-free or mild temperature conditions. Introduction of superplasticity to organic crystals, which can be called "organosuperplasticity," will be an effective strategy to achieving enhanced workability and toughness and exhibited crystalline properties simultaneously.

## Methods
Experimental information is summarized in Supplementary Note 1.

**Preparation of crystals.** N,N-Dimethyl-4-nitroaniline (**1**) was purchased from Tokyo Chemical Industry and acetone was purchased from Wako. Both were used as received. Well-formed single crystals were prepared by recrystallization of **1** from a concentrated solution in acetone.

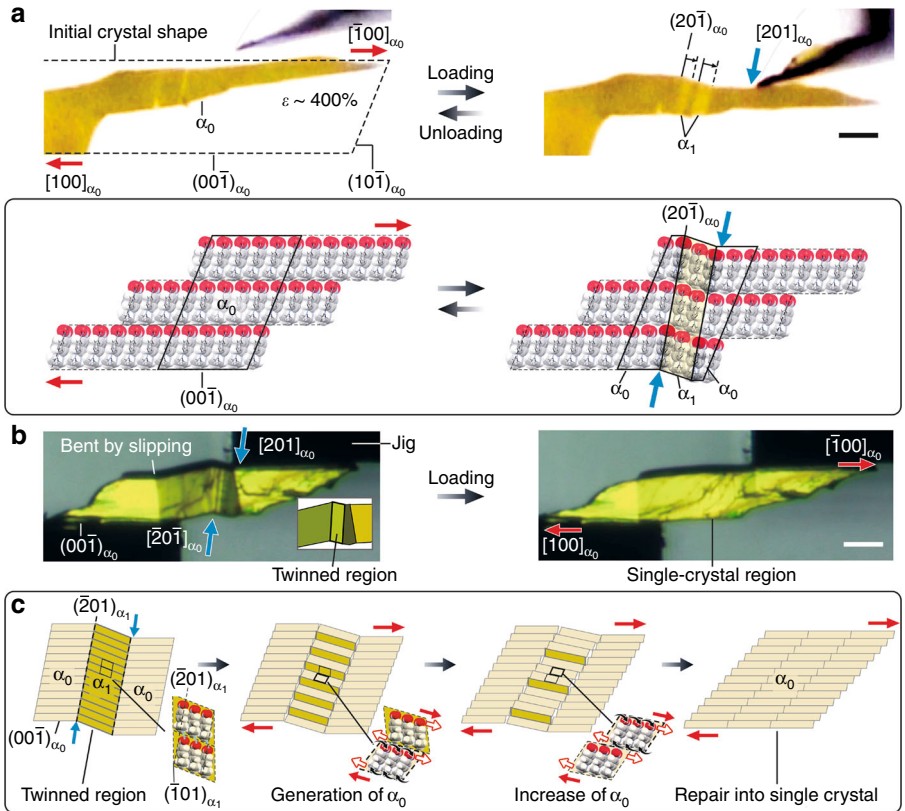

**Fig. 4** Coupling of superplasticity and superelasticity. **a** Superelastic behavior of crystals of **1** after superplastic deformation. The two bright crystal domains ($\alpha_1$) were observed under a polarizing microscope after shear stress loading. The domains disappeared when the stress was removed. Scale bar, 50 μm. The schematic model below the crystal photos represents and explains the superelastic deformation in single crystalline regions. **b** Superplastic behavior during superelastic deformation. Firstly, beam bending by slipping was induced in the crystal of **1** when the jig at the top right of the photo was moved upward. Then the jig was moved downward, twinning a part of the crystal (the bright region in the inset). **c** Schematic model explaining the recovery of single crystallinity by the generation and increase of $\alpha_O$ domains in $\alpha_1$ domains resulting from multi-layer slipping. Scale bar, 200 μm

**Single-crystal X-ray diffraction**. Single-crystal X-ray diffraction data were collected on a Bruker Smart APEX diffractometer equipped with CCD area detector and using graphite-monochromated Mo-Kα radiation ($\lambda = 0.71073$ Å). Data collections were carried out at 298 K. Empirical or multi-scan absorption corrections were applied using SADABS. The structures were solved by direct methods (SHELXS-97 or SHELXT-2014/5) and refined by full-matrix least squares calculations on $F^2$ (SHELXL-97 or SHELXL-2014/7). Non-hydrogen atoms were refined anisotropically, while hydrogen atoms were fixed at calculated positions and refined using a riding model. Crystal face indexing was performed using SMART in a SHELXL Ver. 6.12 program package. Crystallographic data of the structures are summarized in Supplementary Table 1. CCDC 1847179–1847182 and 1858314 contain the supplementary crystallographic data.

**Microscope observations**. A polarization optical microscope coupled with a digital camera was used to record mechanical twinning using tweezers.

**Force measurements**. Shear tests were carried out on a universal testing machine (Tensilon RTG-1210, A&D Co. Ltd.). A shearing speed of 50 μm min$^{-1}$ was used at a constant temperature of 298 K. Conditions and results of shear tests are summarized in Supplementary Note 4 (Supplementary Table 2 and Supplementary Figures 9–11).

## Data availability
All the data generated or analyzed during this study are included in this published article (and its supplementary information files) or available from the authors upon reasonable request. The crystallographic data in this study have been deposited in the Cambridge Structural Database under entry ID CCDC 1847179–1847182 and 1858314.

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

## Acknowledgements

Funding was provided by 2016–2018 Strategic Research Promotion (SK2810) of Yokohama City University for S.T., JSPS KAKENHI Grants JP17H06368 (Grant-in-Aid for Scientific Research on Innovative Areas) and JP17K19143 (Grant-in-Aid for Challenging Research (Pioneering)) for S.T., the SUZUKI Foundation, and the Kanagawa Institute of Industrial Science and Technology.

## Author contributions

S.T. designed the project and edited the manuscript. Y.T. carried out the experiments and analyzed the data. T.S. and N.O. contributed to preparation of the manuscript.

## Additional information

**Competing interests:** The authors declare no competing interests.

