## [Peer Review File · Nature Communications]

Reviewers' comments:

Reviewer #1 (Remarks to the Author):

It is TRUE that organic crystals show difficulties in molding even though they exhibit high flexibility in molecular design. If one can deform materials freely, it would be very useful to mold the materials to products. In this sense, this paper dealing with the first observation of superplasticity in organic crystals, I believe, warrants publication in Nature Communications. However, the authors should address to the following comments satisfactorily before acceptance:

1) I. 30: 'However, has limited application- - -'

Some words are missing.

2) I. 219-222: 'Introduction of superplasticity to organic crystals, which can be called 'organosuperplasticity', - - -'

I agree that introduction of superplasticity to organic crystals will be very useful to provide organic crystals with high modability, but most important and critical is how one can do it. The authors have shown the first example of superplasticity of an organic crystal, compound 1, but only one example. What is the most important issue for organic crystals to exhibit the superplasticity? Crystal structures? Hydrogen bondings? I believe that the authors should suggest the methodology to afford organic crystals the 'superplasticity'.

Reviewer #2 (Remarks to the Author):

This is an extremely interesting paper that reports novel and exciting properties of a compound that has been studied previously by many other researchers, none of whom seem to have noticed these really striking mechanical properties. The videos are very impressive. Taking into account the importance of the conclusions, further experimental data would be very important, in order to understand better the reported phenomenon and to estimate the potential of its practical application:

1. In addition to the videos and plots illustrating macroscopic deformation, one needs plots of at least selected representative 2D X-ray diffraction frames collected from a crystal before the macroscopic deformation, after the deformation (possibly at different stages of deformation), and after the crystal has restored its shape, to evaluate the degree of disorder and strain accumulated. It is not sufficient to deposit only CIFs of initial crystals, one needs also the data that characterise strain and disorder in the crystals before deformation, after it, after shape restoration - in multiple cycles. Data on X-ray diffuse scattering would be informative.

2. A material undergoes multiple cycles of deformation - shape restoration. The paper lacks information on multiple cycles of straining the same crystals - is there any fatigue in macroscopic mechanical properties and in the microscopic defect structure, domains and disorder.

3. Data on indenting the crystals would be informative to characterise mechanical properties.

4. Data on polarized light optical microscopy of the crystal are needed: before deformation, at different stages of deformation, on restoring the original shape, etc - in multiple cycles.

Based on the material presented in the manuscript, I would not call the process a single-crystal to single-crystal transformation without further explanations. The role of twinning, domain structure and its changes in the macroscopic strain is clear. The extent to which the original properties and structure are preserved needs to be clarified by direct experiments.

A few comments are related to references to related phenomena. In addition to mentioning superplasticity of metals and polymers, it would be appropriate to mention also plastic organic crystals (see e.g. <https://phys.org/news/2014-01-plastic-crystals-possibilities-materials.html>),

rotor phases. and liquid crystals with unique plastic properties, like bending and twisting liquid crystals.

Reviewer #3 (Remarks to the Author):

The article written by Satoshi Takamizawa and co-workers reveals the unprecedented example of organic crystalline material that show superplastic property. I am truly impressed of this finding, as common organic molecular crystals are fragile under external stress, and do not undergo elastic or plastic deformations. While superelastic behaviour has been encountered for several low-mass (non-polymeric) organic materials being the matter of current discussions, only some metal alloys demonstrate superplastic properties, and, according to my best knowledge, the stress-induced single-crystal to single-crystal superplastic deformation has never been observed for organic crystals so far. Moreover, as demonstrated in this work, the crystal of N,N-dimethyl-4-nitroaniline can undergo two different types of deformation – i.e. superplastic (parallel slipping) and superelastic (slipping with twinning) depending on the direction of external forces. As demonstrated by authors, the deformation process is regulated by specific rail interlock-type molecular arrangement governed by weak intermolecular hydrogen bond interactions between NMe₂ and NO₂ groups. At this point my question is what happen when shear stress is applied along other crystallographic directions? Does the crystal fracture? Regarding hydrogen bond interaction analysis (p.5 l: 83) it seems that the second N-methyl group is also involved in HB contact with another oxygen atom from neighboured molecule, The C...O contact is only slightly longer ($\sim 3.6 \text{ \AA}$), nevertheless, it provides non-negligible contribution to the total dimer stabilization, and thus it should be described as well. Furthermore, the molecular arrangement also results from the dipole-dipole interactions. In my opinion it would be very interesting to look closer to the bonding situation from the energetic perspective especially in slipping contact region. The theoretical calculations may be very useful in this case and periodic DFT interlayer energy estimations may provide deeper insight into the structure-energy-mechanical property relationship. For instance, the α_0 - α_1 interphase region can be modelled and the interlayer interaction energy compared with the one obtained for pure component, and further related to the results of mechanical strain experiments. Naturally, this can be the matter of another study.

The two-stage superelastic-superplastic experiment show surprising crystal shape recovery which is accompanied by the disappearance of twinned fraction. This was nicely shown on figure 4b and supplemented videos. However, it would be more convincing to confirm the structural recovery by single-crystal X-Ray diffraction experiment.

In my opinion this work will stimulate new studies in the field of organic-material chemistry and will change our viewpoint on the properties of organic compounds. Therefore, I fully support publication of this material in Nature Communications.

Krzysztof Durka,
Warsaw University of Technology
Faculty of Chemistry
Noakowskiego 3
00-664 Warsaw
Poland

Responses to reviewer comments for “Superplasticity in an Organic Crystal”

Reviewer #1 (Remarks to the Author):

Thank you for your valuable comments.

It is TRUE that organic crystals show difficulties in molding even though they exhibit high flexibility in molecular design. If one can deform materials freely, it would be very useful to mold the materials to products. In this sense, this paper dealing with the first observation of superplasticity in organic crystals, I believe, warrants publication in Nature Communications.

However, the authors should address to the following comments satisfactorily before acceptance:

1) l. 30: ‘However, has limited application’ - - ‘

Some words are missing.

We have inserted “it” into the sentence: “However, it has limited application...”

2) l. 219-222: ‘Introduction of superplasticity to organic crystals, which can be called ‘organosuperplasticity’, - - ‘

I agree that introduction of superplasticity to organic crystals will be very useful to provide organic crystals with high modability, but most important and critical is how one can do it. The authors have shown the first example of superplasticity of an organic crystal, compound 1, but only one example. What is the most important issue for organic crystals to exhibit the superplasticity? Crystal structures? Hydrogen bondings? I believe that the authors should suggest the methodology to afford organic crystals the ‘superplasticity’.

We think that our observation of organosuperplasticity present an important first step, discussing the discovery of superplasticity in organic crystals, and there is insufficient data to present an appropriate argument about the essential key factors generating this superplasticity. Thus, we concentrated on accurately describing our observations in this paper. We hope the results shown in the paper will contribute to materials design (methodologies) bringing about superelasticity in organic crystals and related materials.

Reviewer #2 (Remarks to the Author):

Thank you for your kind suggestion.

This is an extremely interesting paper that reports novel and exciting properties of a compound that has been studied previously by many other researchers, none of whom seem to have noticed these really striking mechanical properties. The videos are very

impressive. Taking into account the importance of the conclusions, further experimental data would be very important, in order to understand better the reported phenomenon and to estimate the potential of its practical application:

[General response to Reviewer 2] At first, for proof of application, a technique for strictly maintaining the single-crystal structure should be given top priority in this new materials science discipline. The potential from the retention of crystallinity can be seen from a scientific perspective in further experiments at the next stage with the addition of non-ideality (in terms of fluctuation, stress gradient, structural gradient, macro-phases, micro-phases, domain multiplicity, intermediation, transition, migration, etc.) to organosuperplastic materials. Since no study so far has elucidated the superplastic materials to which the single-crystal techniques can be applied, we concentrated on experimental observations under crystal conditions that were kept as ideal as possible in this paper.

At present, I think we have not reached a stage where it is feasible to present sound arguments about potential applications.

1. In addition to the videos and plots illustrating macroscopic deformation, (1) one needs plots of at least selected representative 2D X-ray diffraction frames collected from a crystal before the macroscopic deformation, after the deformation (possibly at different stages of deformation), and after the crystal has restored its shape, (1') to evaluate the degree of disorder and strain accumulated. It is not sufficient to deposit only CIFs of initial crystals, one needs also the data that characterise strain and disorder in the crystals before deformation, after it, after shape restoration – (2) in multiple cycles. Data on X-ray diffuse scattering would be informative.

(1 and 1') We have added information related to 2D X-ray diffraction frames in SI to examine data on diffraction at different deformation rates on superplastic deformation (Table S1, addition of CCDC1858314).

The following is the additional text on page 6 in SI:

(Retention of crystallinity)

The crystals of **1** maintain their crystallinity during not only the superplastic deformation but also the superelastic twinning. As-prepared, superelastically twinned, and superplastically deformed crystals of **1** were examined by viewing the raw X-ray diffraction data (Figs. S3–S6) converted to reciprocal space. The superelastically twinned crystals of **1** exhibited merged reflections of mother and daughter domains. (Fig. S4b(ii)). The superplastically deformed crystals of **1** exhibited ideal reflections,

which were essentially identical to those of the as-prepared crystals (Fig. S5 and S6), while a certain tailing at some spots was eventually observed in the largely deformed crystals (deformation rate of 330%) through our experimental techniques. The observed manner of tailing of the spots on the reflections from the (h0l) plane in the raw diffraction data is most likely due to a slight deflection in the <001> direction of the stretched thin crystal specimen (Fig. S6b(ii)).

We thus concluded that the crystals of **1** maintain their crystallinity during the superplastic deformation as well as during the superelastic twinning.

(2) Even though we believe our current experimental techniques are excellent, the best at present, it would be hard for us to examine multiple-cycle test results with these techniques, especially when the results include superplastic (slipping) deformation.

2. A material undergoes multiple cycles of deformation - shape restoration. The paper lacks information on multiple cycles of straining the same crystals - is there any fatigue in macroscopic mechanical properties and in the microscopic defect structure, domains and disorder.

I apology for our inability to appropriately provide your requested data at present for the reason given in our answer in (2) above. However, we believe the current data are moderately sufficient for a satisfactory explanation of novel properties in the discussion and conclusion in the paper.

3. Data on indenting the crystals would be informative to characterise mechanical properties.

We did not perform an experiment on the crystals actively distorted due to spatial stress gradients in this study. Although experiments on indentation of the crystals, including nano-indentation, could provide some fruitful knowledge, such experiments would clarify the secondary characteristics of crystals discussed in this paper. The primary focus should be to present knowledge confirming the fundamental findings, including basically how our conventional techniques preserve crystallinity during both plastic and twinning crystal deformations. (Please refer to the [General response to Reviewer 2] as well.)

4. Data on polarized light optical microscopy of the crystal are needed: before deformation, at different stages of deformation, on restoring the original shape, etc - in multiple cycles.

We used a light polarization technique appropriately to record the crystal deformations in this paper. Information has been added/modified in the Methods in the paper and the experimental information in SI. A description about the polarization technique has been left in each figure caption in the paper.

Based on the material presented in the manuscript, I would not call the process a single-crystal to single-crystal transformation without further explanations. The role of twinning, domain structure and its changes in the macroscopic strain is clear. The extent to which the original properties and structure are preserved needs to be clarified by direct experiments.

Indeed, it is hard to precisely define the state of a single crystal. From various points of view, all so-called single crystals are not perfectly single crystalline. The term “single-crystal to single-crystal” (SCSC) has been used for the transition between single crystals, which are considered as single crystals by conventional experiments such as X-ray diffraction. Since we could apply conventional single-crystal X-ray diffraction analysis to the specimens in this paper, we believe that using “SCSC” give readers a better understanding of the material. I would like to emphasize that the observed behavior in this paper is considerably SCSC because each single-crystal (macro) domain is connected by an extremely small volume of planar boundary zones, which are not much recognizable as crystals. While we understand your concerns about the manner of interpretation of “SCSC” during deformation possibly being affected by complex factors, we believe that our observations cover the primary behaviors to provide an understanding of the main phenomena of organosuperplasticity.

A few comments are related to references to related phenomena. In addition to mentioning superplasticity of metals and polymers, it would be appropriate to mention also plastic organic crystals (see e.g. <https://phys.org/news/2014-01-plastic-crystals-possibilities-materials.html>), rotor phases. and liquid crystals with unique plastic properties, like bending and twisting liquid crystals.

We heartily thank you for letting us know about the interesting behaviors in plastic crystals. However, we did not find an essential similarity in the properties of plastic crystals that you pointed out to us. There is no necessity to refer to the properties of plastic crystals in this paper. The mesophases (plastic crystals or liquid crystals) have a lower ordered condensed state than the three-dimensionally ordered state of (single) crystals. The meaning of the word “plastic” in plastic crystals is completely different

from the present “plastic” property in the physical sense. (We will carefully examine the contents of the papers related to the interesting behaviors of mesophases later and may cite them in our next paper if scientifically applicable.)

Reviewer #3 (Remarks to the Author):

Thank you for your kind and valuable comments.

The article written by Satoshi Takamizawa and co-workers reveals the unprecedented example of organic crystalline material that show superplastic property. I am truly impressed of this finding, as common organic molecular crystals are fragile under external stress, and do not undergo elastic or plastic deformations. While superelastic behaviour has been encountered for several low-mass (non-polymeric) organic materials being the matter of current discussions, only some metal alloys demonstrate superplastic properties, and, according to my best knowledge, the stress-induced single-crystal to single-crystal superplastic deformation has never been observed for organic crystals so far. Moreover, as demonstrated in this work, the crystal of N,N-dimethyl-4-nitroaniline can undergo two different types of deformation – i.e. superplastic (parallel slipping) and superelastic (slipping with twinning) depending on the direction of external forces. As demonstrated by authors, the deformation process is regulated by specific rail interlock-type molecular arrangement governed by weak intermolecular hydrogen bond interactions between NMe₂ and NO₂ groups. (1) At this point my question is what happen when shear stress is applied along other crystallographic directions? Does the crystal fracture? (2) Regarding hydrogen bond interaction analysis (p.5 l: 83) it seems that the second N-methyl group is also involved in HB contact with another oxygen atom from neighboured molecule. The C...O contact is only slightly longer (~3.6 Å), nevertheless, it provides non-negligible contribution to the total dimer stabilization, and thus it should be described as well. Furthermore, the molecular arrangement also results from the dipole-dipole interactions. In my opinion it would be very interesting to look closer to the bonding situation from the energetic perspective especially in slipping contact region. (3) The theoretical calculations may be very useful in this case and periodic DFT interlayer energy estimations may provide deeper insight into the structure-energy-mechanical property relationship. For instance, the $\alpha 0$ - $\alpha 1$ interphase region can be modelled and the interlayer interaction energy compared with the one obtained for pure component, and further related to the results of mechanical strain experiments. Naturally, this can be the matter of another study.

(1) The crystal curves or is cleaved if shear stress is actively applied on the crystal in a different direction from the easy-shear directions of twinning and sliding deformations.

The crystals in this paper possess a certain strong tenacity. We confirmed superelastic behavior even in the distorted crystals. (Please see Video S2a.)

(2) As you have described, the length for the alternative $N = O \cdots H-C$ hydrogen bond (3.610 Å) has been added to the main text: “ $N = O \cdots H-C$ hydrogen bonds (3.588 and 3.610 Å from the oxygen to carbon atoms).” Accordingly, in Fig. 1c, the blue line indicating a hydrogen bond has been redrawn as double lines.

(3) We believe that we have reached a sufficiently worthwhile conclusion with the analysis of the current experimental data without discussing theoretical calculations and scientific models in this paper. (We understand the power of DFT calculations for further examination. We hope to report some sort of results associated with such calculations elsewhere in the near future.)

The two-stage superelastic-superplastic experiment show surprising crystal shape recovery which is accompanied by the disappearance of twinned fraction. This was nicely shown on figure 4b and supplemented videos. However, it would be more convincing to confirm the structural recovery by single-crystal X-Ray diffraction experiment.

Due to the separation of the deformation stage from the X-ray diffractometer, it is difficult to carry out reasonable single-crystal X-ray diffraction experiments on the crystals after multiple treatments with our present techniques. (Please refer to [General response to Reviewer 2] to understand our position at the current stage.) Fortunately, the observations with an optical microscope during deformation are extremely effective in this paper. (Going forward, we are making efforts to develop our observation techniques toward *in-situ* X-ray diffraction measurements in this research.)

In my opinion this work will stimulate new studies in the field of organic-material chemistry and will change our viewpoint on the properties of organic compounds. Therefore, I fully support publication of this material in Nature Communications.

REVIEWERS' COMMENTS:

Reviewer #1 (Remarks to the Author):

As far as my comments are concerned, the authors have addressed them reasonably. Therefore, I recommend publication of this revised manuscript in Nature Commun.

Reviewer #2 (Remarks to the Author):

The authors have addressed some of the comments from my review and from the review of the Reviewer 3 in their revised version. I keep the opinion that the additional studies would be important, in order to understand to which extent the reported phenomenon is unique, and potentially practically important. For example, knowing how a material behaves on multiple cycling and if there is a fatigue phenomenon is crucially important if any real applications are foreseen. I also remain convinced that even if the authors use the term "plastic crystal" in a different meaning than the one used before in the literature, this is not a reason not to refer to earlier published work. On the contrary, they should note that the same or similar term has been used before to describe a different phenomenon (or the phenomenon they consider to be different). I understand the response concerning using the term "single crystal to single crystal transformation", but I am still confident that at least an explanation of what the authors define like a single crystal to single crystal transformation is absolutely needed in the text of the paper. A true single crystal and a multi-domain or twinned crystal is not the same. Twinning resulting from a mechanical action is a very common phenomenon, whereas preserving a true single crystal even after multiple cycles of reversible large deformation would be a really interesting phenomenon. However, I still do not see a proof of the observation of a truly unique phenomenon in this paper.

That said, I do not mind if the material is published: the observations remain valid, even though their interpretation can be disputable.

I would however recommend - I do not insist on this, since these are the authors and not me as a reviewer, who are at the final end responsible for what is published,- that the authors use the terminology more carefully and do not try to ascribe to their observations more than is in fact known and proved at this stage. I would find it to be a good idea to write what remains NOT proved and NOT properly studied at this stage directly in the paper.

Reviewer #3 (Remarks to the Author):

I have carefully examined the answers and revised version of the manuscript. I am satisfied with the authors corrections, thus I support the publication of presented material. I have only one small remark. Authors did not provide experimental errors for the lengths of hydrogen bond interactions on p.5. These can be added on the next stage of publication procedure.

Krzysztof Durka,
Warsaw University of Technology
Faculty of Chemistry
Noakowskiego 3
00-664 Warsaw
Poland

Responses to reviewer comments for “Superplasticity in an Organic Crystal”

Reviewer #1 (Remarks to the Author):

As far as my comments are concerned, the authors have addressed them reasonably. Therefore, I recommend publication of this revised manuscript in Nature Commun.

Thank you for your valuable comments.

Reviewer #2 (Remarks to the Author):

The authors have addressed some of the comments from my review and from the review of the Reviewer 3 in their revised version. I keep the opinion that the additional studies would be important, in order to understand to which extent the reported phenomenon is unique, and potentially practically important. For example, knowing how a material behaves on multiple cycling and if there is a fatigue phenomenon is crucially important if any real applications are foreseen. I also remain convinced that even if the authors use the term "plastic crystal" in a different meaning than the one used before in the literature, this is not a reason not to refer to earlier published work. On the contrary, they should note that the same or similar term has been used before to describe a different phenomenon (or the phenomenon they consider to be different). I understand the response concerning using the term "single crystal to single crystal transformation", but I am still confident that at least an explanation of what the authors define like a single crystal to single crystal transformation is absolutely needed in the text of the paper. A true single crystal and a multi-domain or twinned crystal is not the same. Twinning resulting from a mechanical action is a very common phenomenon, whereas preserving a true single crystal even after multiple cycles of reversible large deformation would be a really interesting phenomenon. However, I still do not see a proof of the observation of a truly unique phenomenon in this paper.

That said, I do not mind if the material is published: the observations remain valid, even though their interpretation can be disputable.

I would however recommend - I do not insist on this, since these are the authors and not me as a reviewer, who are at the final end responsible for what is published,- that the authors use the terminology more carefully and do not try to ascribe to their observations more than is in fact known and proved at this stage. I would find it to be a good idea to write what remains NOT proved and NOT properly studied at this stage

directly in the paper.

We heartily thank you for letting us know about your opinion again. We had carefully responded to your opinion in the previous revision.

Reviewer #3 (Remarks to the Author):

I have carefully examined the answers and revised version of the manuscript. I am satisfied with the authors corrections, thus I support the publication of presented material. I have only one small remark. Authors did not provide experimental errors for the lengths of hydrogen bond interactions on p.5. These can be added on the next stage of publication procedure.

Thank you for your kind comments.

In the parts for H-bond distances, we have replaced “3.588 and 3.610 Å” with “*ca.* 3.6 Å” in the main text (P5L80) and added the precise distances with experimental errors in Table S1 in SI (p5) besides.